# Piezoelectric Response in Hybrid Micropillar Arrays of Poly(Vinylidene Fluoride) and Reduced Graphene Oxide

**DOI:** 10.3390/polym11061065

**Published:** 2019-06-20

**Authors:** Igor O. Pariy, Anna A. Ivanova, Vladimir V. Shvartsman, Doru C. Lupascu, Gleb B. Sukhorukov, Tim Ludwig, Ausrine Bartasyte, Sanjay Mathur, Maria A. Surmeneva, Roman A. Surmenev

**Affiliations:** 1Physical Materials Science and Composite Materials Centre, National Research Tomsk Polytechnic University, 634050 Tomsk, Russia; igor-parij1995@mail.ru (I.O.P.); metallurg_annet@mail.ru (A.A.I.); surmenevamaria@mail.ru (M.A.S.); 2Institute for Materials Science and Center for Nanointegration Duisburg-Essen (CENIDE), University of Duisburg-Essen, 45141 Essen, Germany; vladimir.shvartsman@uni-due.de (V.V.S.); doru.lupascu@uni-due.de (D.C.L.); 3School of Engineering and Materials Science, Queen Mary University of London, London E1 4NS, UK; g.sukhorukov@qmul.ac.uk; 4University of Cologne, 50923 Cologne, Germany; tim.ludwig@uni-koeln.de (T.L.); sanjay.mathur@uni-koeln.de (S.M.); 5FEMTO-ST Institute, 25000 Besançon, France; ausrine.bartasyte@femto-st.fr

**Keywords:** piezoelectric response, reduced graphene oxide, poly(vinylidene fluoride), hybrid film

## Abstract

This study was dedicated to the investigation of poly(vinylidene fluoride) (PVDF) micropillar arrays obtained by soft lithography followed by phase inversion at a low temperature. Reduced graphene oxide (rGO) was incorporated into the PVDF as a nucleating filler. The piezoelectric properties of the PVDF-rGO composite micropillars were explored via piezo-response force microscopy (PFM). Fourier transform infrared spectroscopy (FTIR) and X-ray diffraction (XRD) showed that α, β, and γ phases co-existed in all studied samples, with a predominance of the γ phase. The piezoresponse force microscopy (PFM) data provided the local piezoelectric response of the PVDF micropillars, which exhibited a temperature-induced downward dipole orientation in the pristine PVDF micropillars. The addition of rGO into the PVDF matrix resulted in a change in the preferred polarization direction, and the piezo-response phase angle changed from −120° to 20°–40°. The pristine PVDF and PVDF loaded with 0.1 wt % of rGO after low-temperature quenching were found to possess a piezoelectric response of 86 and 87 pm/V respectively, which are significantly higher than the |d_33_^eff^| in the case of imprinted PVDF 64 pm/V. Thus, the addition of rGO significantly affected the domain orientation (polarization) while quenching increased the piezoelectric response.

## 1. Introduction

Poly(vinylidene fluoride) (PVDF) is a well-known ferroelectric polymer with many promising properties and applications in micro electro mechanical systems (MEMS). For instance, the low elastic modulus and acoustic impedance make PVDF applicable as transducers in water, organic materials, and biological tissue [1,2]. Due to the moderate piezoelectric voltage requirements and polymorphism in PVDF, which determine the physical and electromechanical performance of the polymer, there is growing interest in the development of MEMS based on the PVDF used mostly in hydrophones [3], in ultrasonic sensors [4,5], as pressure sensors [6], and as energy harvesters [7]. A great deal of research on the crystallization of the β phase in PVDF has been performed, which has superior piezoelectric properties to the γ phase. The β and/or γ crystals in PVDF could be obtained from the non-polar α modification by various processes, such as stretching [8,9], poling under high electric fields [10,11], melt crystallization under high pressure [12,13], or very high cooling rates [14,15,16]. While PVDF films formed by conventional methods, such as melt casting and spin coating, primarily consist of the α-phase [17], those obtained by phase inversion are dominated by the β-phase [16].

In addition to the processing conditions to tune the various polymorphs, patterned PVDF micro- and nano-structures have attracted a great deal of attention for the development of MEMS. Several approaches have been used to pattern PVDF micro- and nano-structures, including electron/ion beam writing [18,19], layer-by-layer assembly [20], nanoimprinting [21], and soft lithography [22].

It has been shown that nano- or micro-confinement fosters the self-orientation of ferroelectric crystallites in PVDF and promotes the formation of electroactive polymorphs [23,24,25,26]. Ong et al. observed that the piezo-response of PVDF films becomes enhanced with a reduction of feature size, probably leading to confined crystallization inducing polymer chain alignment, consequently resulting in enhanced piezoelectric phase content [27]. In addition, it was shown that micropatterning resulted in the improved piezo-response by 1.85 times from 40(2) to 74(2) pm/V, in comparison with the thin film counterpart due to the geometrical strain confinement effect in P(VDF-TrFE) core-shell structures [28].

Alternative strategies to affect the polymorph phase and the electro-mechanical properties of PVDF involve the addition of secondary phases as nanofillers into the polymer matrix, which act as nucleating agents that stabilize the polar β-phase. Graphene oxide (GO) and its derivatives are often used to improve the energy-harvesting performance of PVDF, and it has been shown that GO contributes to the alignment of dipoles in PVDF and the enhancement of the dielectric and piezoelectric charge constant values due to the formation of micro-capacitors inside the composite [29,30,31].

This report demonstrates the synergistic effect of unifying the above-mentioned approaches by (i) patterning the PVDF followed by (ii) low-temperature quenching and (iii) adding nanofiller with the goal of driving the molecular organization to the electroactive phase and improving the piezoelectric performance. We investigated the effects of preparation conditions, namely, low-temperature quenching and concentration of the rGO filler, on the piezoelectric performance of the PVDF micropillar array.

## 2. Experimental

### 2.1. Reduced Graphene Oxide (rGO)

Graphene oxide was prepared from natural graphite by an improved Hummers method [32]. For this, a 9:1 mixture of concentrated H_2_SO_4_/H_3_PO_4_ (360:40 mL) (purity 99.999%, Sigma Aldrich, Steinheim, Germany) was added to a mixture of graphite flakes (3.0 g) (particle size distribution + 100 Mesh, Sigma Aldrich, Steinheim, Germany) and KMnO_4_ (18.0 g) (purity ≥99%, Sigma Aldrich, Steinheim, Germany). The reactants were then heated to 50 °C and stirred for 12 h. The product was allowed to cool to room temperature (RT) and poured onto ice (ca. 400 mL). Thereafter, 30% H_2_O_2_ (3 mL) (30 wt % in H_2_O, Sigma Aldrich, Steinheim, Germany) was added to the product until a clear colour change could be noted. The resulting brownish product was centrifuged (4500 rpm, 15 min) and consecutively washed with water, 30% HCl (30 wt % in H_2_O, Merck KgaA, Darmstadt, Germany), and ethanol (≥99.8%, Sigma Aldrich, Steinheim, Germany). The solid obtained on the filter was vacuum-dried overnight at room temperature. To obtain reduced graphene oxide (rGO), the GO was thermally reduced in a tube furnace at 800 °C in an Ar/H_2_ atmosphere (95/5). The Raman spectrum of the synthesized rGO reveals two prominent peaks in the G band and D band (Figure 1).

The *I*_D_/*I*_G_ intensity ratio of rGO is calculated to be 0.6, which is in agreement with the results reported elsewhere [33].

### 2.2. Micropillar Preparation

The rGO powder was dispersed in N,N-dimethylformamide (DMF, Sigma Aldrich) solvent and ultra-sonicated for 2 h at RT, and then mixed under magnetic stirring for 1 h at 50 °C. PVDF (Sigma Aldrich, M_w_ ~534,000) powder (20 wt %) was separately dispersed in 10 mL DMF under magnetic stirring for 1 h at 50 °C. Then, a rGO/DMF solution with various concentrations of rGO was added to the PVDF/DMF solution to obtain PVDF/rGO solutions of different concentrations (0.1, 0.4 wt % rGO) [34,35,36,37]. The PVDF/rGO dispersion was stirred at 700 rpm at 60 °C for 10 h to achieve a good interaction of rGO with PVDF.

Figure 2 shows a schematic of the fabrication process of the PVDF micropillar arrays. A polydimethylsiloxane (PDMS) stamp replicated from a silicon master featuring pillars (5 μm diameter, 4 μm height and with a period of 20 μm) was used, as displayed in Figure 2. PDMS mixed with a curing agent (10:1, Sylgard 184 from Dow-Corning, Midland, MI, USA) was poured onto the silicon master produced via standard lithography at Shenzhen Semiconductor, Shenzhen, China. Subsequently, the PDMS was degassed for 30 min in a vacuum, followed by curing the PDMS at 70 °C for 3 h and peeling the PDMS off the silicon master.

Imprinted PVDF micropatterns were produced by a dip-coating of Al foil into the polymer solution. The PDMS stamp was then used to imprint the dip-coated PVDF film, followed by drying at 70 °C for 10 min under pressure (200–310 kPa). Thereafter, the PDMS stamp was subsequently peeled away, leaving the patterned film on the substrate [38]. The imprinted micropillars were then immediately immersed in a precipitation bath comprising a mixture of deionized water and glycerol at −20 °C. The as-prepared patterned films were washed thoroughly with water to remove any traces of glycerol and dried overnight in air at RT.

A portion of the samples was exposed to an external electric field (3.5 MV/m) at 90 °C for 1 h for poling. The poled samples were cooled down in an oven under the applied electric field. Thus, six groups of samples were prepared, as described in Table 1.

The morphology of the PVDF micropillar arrays was investigated by scanning electron microscopy (Au-sputtered samples; ESEM Quanta 400, accelerating voltage 20 kV, Thermo Scientific). Differential scanning calorimetry (DSC) was used (DSC 204 F1 Phoenix equipment) at a heating rate of 5 °C·min^−1^. The samples for the DSC studies were cut into small pieces from the centre region of the patterned films and placed into 40 μL aluminium pans. All experiments were performed in a nitrogen atmosphere by heating the samples up to 200 °C. The total crystallinity of samples, Δ*X*_c_, can be calculated by assuming the heat of fusion of 100% crystalline PVDF to be 104.7 J g^−1^ [39]:(1)$$\Delta Xc=\frac{\Delta H_{m}}{\Delta H_{m}^{100}}\times 100\%$$
where, $\Delta H_{m}$ and $\Delta H_{m}^{100}$ are the melting enthalpy of the sample and the melting enthalpy for a 100% crystalline sample, respectively. FTIR measurements were performed at room temperature in an ALPHA Platinum FTIR instrument in attenuated total reflectance (ATR) mode from 4000 to 650 cm^−1^ with a resolution of 3 cm^−1^. The obtained spectra were analysed in accordance with the methodology described in reference [40]. According to the procedure reported in reference [40], the relative fraction of the electroactive β and γ phases (F_EA_) can be quantified as follows [40]:
F_EA_ = *I_EA_*/[(*K*_840_/*K*_763_) *I_763_* + *I_EA_*](2)
where, I_EA_ and I_763_ are the absorbencies at 840 and 763 cm^−1^ respectively, and K_840_ and K_763_ are the absorption coefficients at the respective wavenumbers, 7.7 × 10^4^ and 6.1 × 10^4^ cm^2^ mol^−1^, respectively.

The quantification of the fractions of the individual β and γ -phases can be performed by using the absorbance (peak area or peak height) of the two bands (1275 and 1234 cm^−1^), as demonstrated in Equations (3) and (4), respectively [40]:(3)$$F\;\left(\beta \right)=\;F_{EA}\times \left(\frac{\Delta H_{\beta^{\prime }}}{\Delta H_{\beta^{\prime }}+\Delta H_{\gamma^{\prime }}}\right)\times 100\%$$
(4)$$\left(\gamma \right)=\;F_{EA}\times \left(\frac{\Delta H_{\gamma^{\prime }}}{\Delta H_{\beta^{\prime }}+\Delta H_{\gamma^{\prime }}}\right)\times 100\%$$
where, $\Delta H_{\beta^{\prime }}$ and $\Delta H_{\gamma^{\prime }}$ are the height differences (absorbance differences) between the peak around 1275 cm^−1^ and the nearest valley around 1260 cm^−1^, and the peak around 1234 cm^−1^ and the nearest valley around 1225 cm^−1^, respectively.

X-ray diffraction analysis was performed using a series D8 Advance Bruker diffractometer with CuKα X-ray radiation. Imaging was carried out in the form of θ–2θ scans (Bragg–Brentano configuration) with generator current voltages of 40 mA and 40 kV, respectively. The 2θ step size was 0.03°, and the measurements were conducted within the range of 10° to 30°.

Topography and piezo-response force microscopy (PFM) measurements were performed using an atomic force microscope Ntegra Prima (NT-MDT, Russia). The measurements were carried out using Pt-coated cantilevers, namely, NSG03/TiN (resonance frequency 90 kHz, force constant 1.74 N m^−1^) and FMG01 (resonance frequency 70 kHz, force constant 3 N·m^−1^), purchased from NT-MDT (Russia). Standard topography mapping with the resolution of 512 × 512 points was carried out in the tapping mode. The piezoresponse force microscopy (PFM) was performed using NSG30/Pt cantilevers in contact mode. The PFM measurements were carried out at a frequency of 734 kHz with a 2.5–10 V amplitude of the a.c. driving signal. To avoid any artifacts on atomic force microscopy (AFM) images, the scanning speed was 0.5 Hz, and set point 1 nA.

The PFM signal averaged over three different locations on one sample was used to compute the average value of the piezo-response amplitude. The vertical piezo-response was calibrated using the deflection sensitivity of the cantilever obtained from the force-displacement curve.

## 3. Results and Discussion

According to SEM data, the fabricated micropillar arrays were uniformly imprinted over a large area (0.8 mm × 0.8 mm) with very few defects, even at the edges of the columns or gratings (Figure 3). Some variation between feature sizes in the PDMS mould and imprinted patterns was observed, which was attributed to the high viscosity of the applied PVDF solution, material shrinkage, and the hydrophobic surface of the PDMS mould.

The applied printing process yielded micropatterns with a height of 2(0.2) μm, a period of 19.5(1.5) μm, and an average width of 9.5(0.2) μm, as determined by the three-dimensional and two-dimensional AFM scans and line profiles depicted in Figure 4.

The addition of rGO and low-temperature quenching did not cause any noticeable microstructural changes (Figure 5). The thickness of the PVDF patterned films (4.0(0.2) μm) was measured by scanning the edge of the film by AFM.

DSC measurements performed to determine the crystal structure and melting behaviour of the samples fabricated with 0.1 wt % and 0.4 wt % of rGO showed comparable endothermic peaks with a maximum at approximately 171 °C (Figure 6).

Deconvolution of the DSC curves showed that the obtained signals were a superposition of three or four peaks. The melting events in the DSC thermogram could be ascribed to the melting of the α and β phases present in the films [15,16,41]. The DSC peaks depended on the heating rate, and the appearance of the endotherms with different melting points was due to the reorganization of crystals on heating [39]. Moreover, the DSC features not only depended on the crystalline phase but were also affected by crystalline defects, which were particularly enhanced by the presence of fillers in polymer composites. There is no consensus in the literature on the melting point of different phases, especially when a filler is added to the PVDF [40]. On the other hand, α and β crystallites have a similar melting point, thus DSC cannot be used to determine these two phases, but only to calculate the degree of crystallinity of the sample (Equation (1)). It can be seen from Figure 6 that imprinted, quenched samples and samples with 0.4 wt % of rGO have similar melting points of different phases. At the same time, the sample with 0.1 wt % of rGO has different melting points of α and β-phases compared to other studied samples, while having the highest degree of crystallinity among imprinted, quenched samples and samples with 0.4 wt % of rGO. The imprinted samples do not reveal similar performance compared with the samples with 0.1 wt % of rGO since higher crystallinity and higher density provide such advantages as low moisture permeability and higher mechanical strength [41]. Due to higher crystallinity, the samples with 0.1 wt % of rGO have better chemical resistance, stiffness, and strength than imprinted, quenched samples and the ones with 0.4 wt % of rGO. The calculated values of crystallinity (Table 2) are much lower than the data published for pristine PVDF films and composites. The crystallinity of PVDF samples obtained by solution-casting, spin-coating, stretching, and quenching was found to be in the range of 40–65% [15,40,42]. However, for arrays of isolated PVDF γ-type nanorods, a crystallinity of 20% was observed by XRD in the work of Garcia-Gutierrez et al. [43].

The crystalline structure of the PVDF micropillar arrays was analysed with FTIR (Figure 7) and X-ray diffraction (Figure 8). For the studied samples, the typical peaks attributed to the *α* phase appeared at 764, 796, 1210, and 1423 cm^−1^. The IR-spectra were processed using the Fourier Self-Deconvolution add-on for the OriginPro 2018. The representative peaks for a planar zigzag β phase appeared at 841 and 1273 cm^−1^ and were assigned to the –CH_2_– wagging vibration and the –CF_2_– symmetric stretching, respectively. The peak that appeared at 1233 cm^−^^1^ is the signature of the electroactive γ phase. The β-phase content present in the PVDF micropillar arrays was calculated from the absorption bands at 764 and 840 cm^−1^, characteristic of the α and β phases respectively, following the procedure reported in reference [40]. The relative amount of β-PVDF according to FTIR data was calculated and is presented in Table 3.

The samples with 0.1 wt % of rGO have the largest amount of β-phase than the imprinted and quenched samples and the samples with 0.4 wt % of rGO. At the same time, the samples with 0.1 wt % of rGO with the largest degree of crystallinity have the least amount of α-phase (thus, the largest amount of electroactive phase) compared to imprinted, quenched samples and the ones with 0.4 wt % of rGO. While the quenched samples, having the lowest degree of crystallinity, have larger amount of β-phase than the imprinted samples, the latter group has a degree of crystallinity greater than the quenched one. A similar discrepancy between the degree of crystallinity and the amount of β-phase has already been observed elsewhere [44]. The authors concluded that the degree of crystallinity did not significantly affect the amount of β-phase and piezoelectric response.

XRD analysis of the PVDF micropillar arrays confirms that reflections from the three crystalline phases of PVDF are present. The XRD peaks at 17.7°, 18.4°, and 20.2° were assigned to the (110), (020), and (021) reflections of the *α*-phase [45]. The peak at 20.7° from the (110) and (200) reflections of the *β*-PVDF crystal plane overlapped with the peak at 20.3°, which was assigned to the (110) reflection of the *γ*-PVDF [45]. The line at 17.05° corresponds to a reflection from the substrate (Figure 8).

The piezoelectric response of the PVDF micropillar arrays was evaluated using PFM. To evaluate the average piezoelectric properties of the samples, we performed PFM measurements on at least two different regions for two samples in each group. Figure 9 depicts the three-dimensional AFM images of individual micropillars and the two-dimensional images of the PFM amplitude and phase signal within a scanning area of 20 × 20 μm^2^. The measurements were performed at a low driving voltage (V_ac_ = 2.5 V), corresponding to an electric field far below the coercive field of PVDF. The phase contrast from the PFM measurements reflects the domain polarity in different sample locations, and the local piezoelectric coefficient of the sample can be extracted from the magnitude of the amplitude signal. Both the amplitude and the phase showed no coupling with the surface topography in any sample.

The vertical amplitude was used in the microscope to record the current output from a photodiode and was calibrated to convert the obtained values to pm. The cantilever oscillation amplitude was calibrated by multiplying the measured amplitude on the photodiode with the inverse of the optical cantilever sensitivity [46].

It should be noted that PFM is a highly efficient method to measure local piezoelectric response only in the case of stiff materials with relatively high Q-factor of the system [47], but polymeric materials are not sufficiently stiff for this. In the case of the polymer films, the quantitative determination of the piezoelectric coefficient d_33_ by PFM represents a longstanding challenge. Signals that do not depend exclusively on the piezoeffect but on other physical effects (long-range electrostatic interactions, non-local interactions between the tip and the sample surface and unknown substrate bending effect) can contribute to the detected PFM response. For a full PFM quantification, proper calibration is necessary [47]. Therefore, the PFM study reported in the current work only provides a qualitative description of the distribution of the piezoelectric effect on the surface.

To quantify the PFM images, statistical distributions of the PFM phase and amplitude values were analysed. Figure 10 shows the resulting histograms.

The image shows a rather wide phase distribution of the imprinted and quenched samples, which was attributed to the presence of domains with oriented polarization with some spatial spread. The preferred domain orientation can be distinguished for the samples containing rGO. For the quenched neat PVDF micropillar arrays, the phase signal exhibits an intense peak at approximately −120° that corresponds to the negatively polarized PVDF. With the addition of the rGO in the PVDF matrix, the phase changed from −120° to 20°–40°, which suggests that the domains were rebuilt in the direction of polarization. This effect is observed only with the addition of a large amount of rGO (0.4 wt %); with 0.1 wt % of the rGO, the orientation of the domains did not change. In addition, Figure 10 shows that the quenched sample and the samples with rGO have an increased amplitude compared with the imprinted sample. An increase in amplitude is a consequence of an improvement in the piezoelectric characteristics and an increase in the piezoelectric coefficient d_33_ [48,49].

A histogram of the amplitude response of the imprinted PVDF micropillars shows a wide range of values from 5 to 200 pm. For the quenched samples and samples with rGO, the values of the amplitude increased, indicating that the low-temperature quenching and incorporation of the rGO enhanced the piezoelectric properties of the PVDF micropillars.

Figure 11 shows the average values of displacement obtained from all measurements as a function of the amplitude with the applied ac voltage. The effective longitudinal piezoelectric coefficient |d_33_^eff^| was quantified by the slope of the given linear equation. The maximum piezoelectricity of the developed PVDF-patterned films was 86 and 87 pm/V for the quenched neat PVDF sample, and one filled with 0.1 wt % of rGO, respectively. The obtained results are summarized in Table 4. These results are 2–3-fold higher than the |d_33_| values (17.5–34 pm/V (pC/N)) observed for PVDF samples obtained by mechanical stretching and/or high electric field poling [50,51] but are quite comparable with the values published for quenched and confined PVDF (Table 4) [16,49,50,51,52].

For instance, Soin el al. studied PVDF films prepared by varying the quenching temperature from 100 to −20 °C and established that the d_33_ value increases with decreasing quenching temperature [16]. The d_33_ value of the PVDF film quenched at −20 °C was measured by PFM to be 49.6 pm/V, which originated from the pure β-phase content (~100%) and high crystallinity of 56%. In addition to quenching, microconfinement may also contribute to the enhancement of the piezoelectric response. For instance, Canavese et al. recorded a higher value of the d_33_ coefficient for PVDF-TrFE micropillars (18 pm/V) compared to that of the flat film (14 pm/V) [26]. The authors attributed this effect to the high degree of crystallinity of the pillars due to the microconfinement. A d_33_ value of 58.5 pm/V was obtained for the PVDF nanoribbon with high γ-phase content and 75% crystallinity [52], comparable to our study. A composite spin-coated film of the copolymer P(VDF-TrFE) with graphene oxide (GO) was studied by Silibin et al. [53]. The experimental data were compared with the simulation outputs, and the piezoelectric response of the films was found to depend on the presence of GO. Simulation of the sandwich structure of PVDF/GO results in a calculated value of 29.8 pm/V for d_33_ [53].

Moreover, the results show that the presence of the electroactive phase, as revealed by the FTIR analysis, and the degree of crystallinity, calculated from the DSC data, are not directly correlated to a high value of the piezoelectric coefficient observed for the samples studied in this work. According to Martin [54] and Steinhart [55], a strong deviation of the crystal structure of PVDF embedded within the porous anodic aluminium oxide template was observed compared to that of the residual film. It was assumed that, similar to the anodic aluminium oxide mould, the interaction of the PDMS walls with the PVDF during printing plays a determinant role in the crystal orientation of the polymer. Directed crystal growth occurs preferentially in the direction of the long axis of the micropillars. García-Gutiérrez et al. produced arrays of isolated ferroelectric γ-type PVDF nanorods connected by a paraelectric supporting film using nanoporous alumina templates [43]. On the interface of the residual film, a phase transition may also be induced by the crystallization of PVDF due to the interaction with the surface of the PDMS stamp. Subsequent quenching leads to a less crystalline polymer in the bulk of the residual film. Thus, the low crystallinity of the prepared PVDF-patterned films calculated by DSC is probably the average value of crystallinity between the micropillars and the residual film [56].

The PFM study performed here provided a qualitative understanding of the enhancement of the piezoelectric response of the PVDF micropillar arrays quenched at −20 °C and embedded with rGO. The piezoelectric performance of the quenched PVDF micropillars and composite micropillars containing 0.1 wt % of rGO yielded the best results of the samples considered. The experiments showed that high-performance PVDF micropillar arrays can be obtained without additional poling at a high voltage.

## 4. Conclusions

The results of this study provided a qualitative indication of the enhancement of the piezoelectric response of PVDF micropillar arrays loaded with rGO and quenched at −20 °C. The obtained PFM data displayed that the maximum piezoelectricity of the developed PVDF-patterned films was 86 and 87 pm/V for the quenched neat PVDF sample and the one loaded with 0.1 wt % of rGO respectively, which is substantially higher than a |d_33_^eff^| of 64 pm/V for the imprinted PVDF. The presence of 0.1 wt % of rGO increased the degree of crystallinity by 15% and decreased the α-phase content compared with neat PVDF. The addition of rGO into the PVDF matrix resulted in a change in the preferred polarization direction, and the piezo-response phase angle changed from −120° to 20°–40°. Piezoelectricity was induced by the domain orientation alignment, which was induced in the PVDF micropillars without the application of a strain or an electric field during the synthesis. The samples with 0.1 wt % of rGO has the highest degree of crystallinity, the largest amount of electroactive phase, and the highest piezoelectric coefficient |d_33_^eff^| among all of the studied samples. The approach used in this work enables one-step printing of ferroelectric patterns without any harsh post-processing steps, which could degrade the functional properties of PVDF. Thus, the addition of rGO significantly affected the domain orientation (polarization) while quenching increased the piezoelectric response. To gain insight into the underlying mechanisms of the observed piezo-response of the patterned PVDF/rGO films, further work will be devoted to investigations of dielectric properties and energy-harvesting performance of the quenched composite PVDF/rGO micropillar arrays. Thus, the fabricated piezoelectric PVDF micropillars reported in this paper can be used in different applications, including tissue engineering scaffolds, vital sign transducers, biomedical energy harvesters, and dynamic sensors at the cellular and subcellular levels.

## Figures and Tables

**Figure 1 polymers-11-01065-f001:**
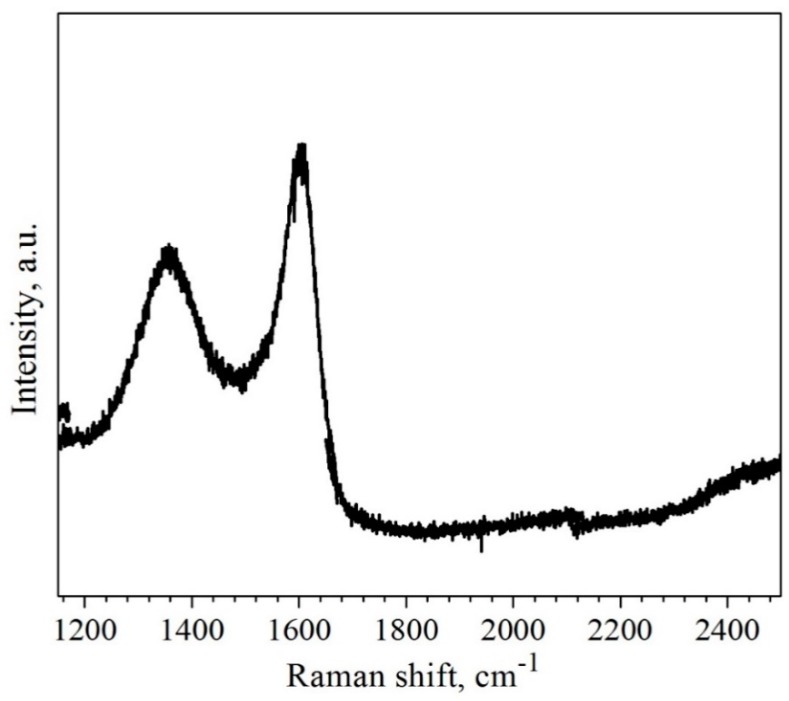
Raman spectrum of the synthesized reduced graphene oxide (rGO) powder.

**Figure 2 polymers-11-01065-f002:**
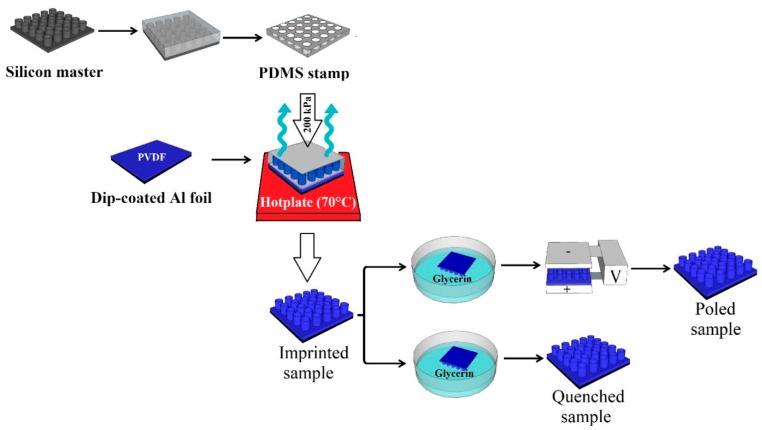
Schematic representation of the preparation of the polydimethylsiloxane (PDMS) soft mould and reverse micro imprint lithography.

**Figure 3 polymers-11-01065-f003:**
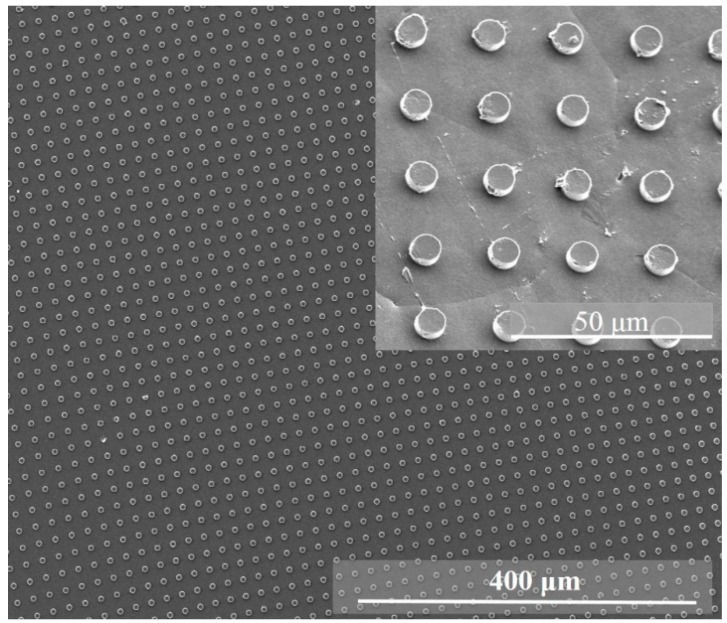
Scanning electron microscopy (SEM) image of the imprinted PVDF micropillar arrays. As an inset an image with a higher magnification is presented.

**Figure 4 polymers-11-01065-f004:**
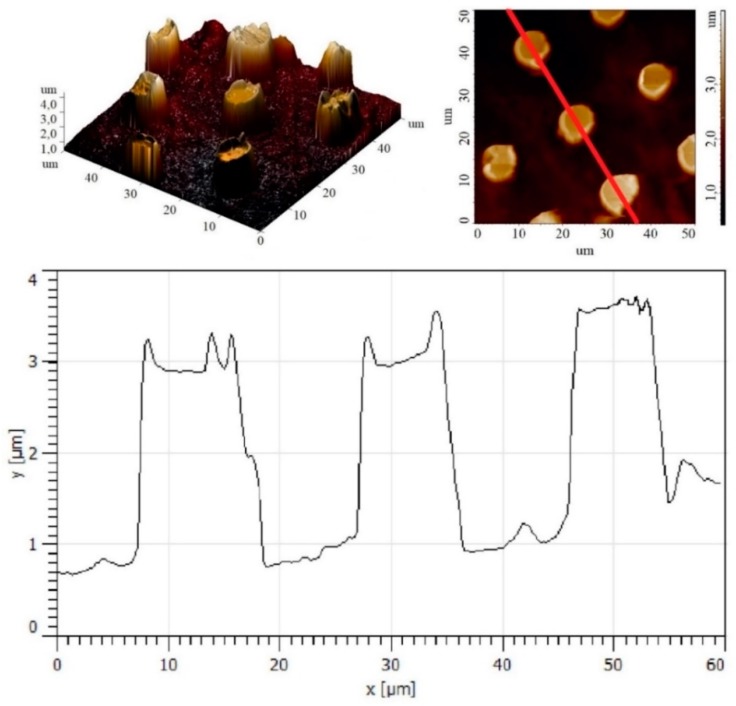
AFM topography of the imprinted PVDF micropillar array and a line scan profile along the line shown on the two-dimensional (2D) topography image.

**Figure 5 polymers-11-01065-f005:**
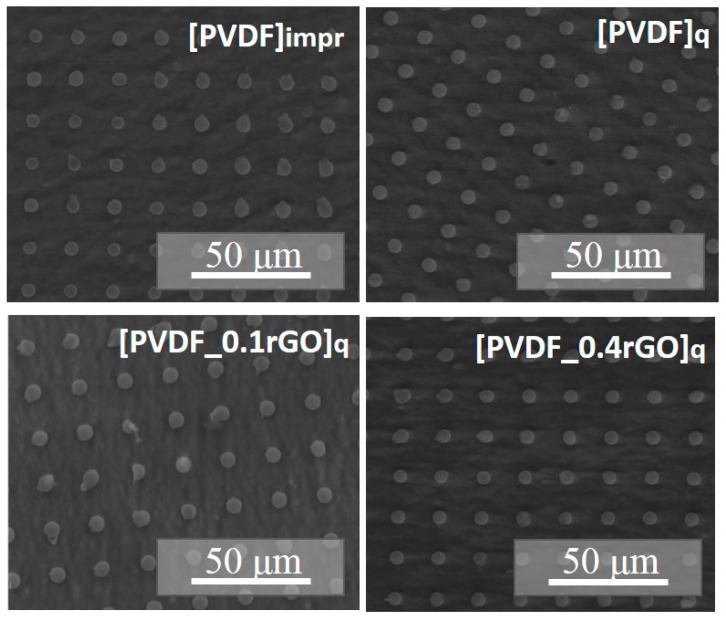
SEM images of the PVDF micropillar arrays imprinted and quenched in glycerol with and without the rGO filler.

**Figure 6 polymers-11-01065-f006:**
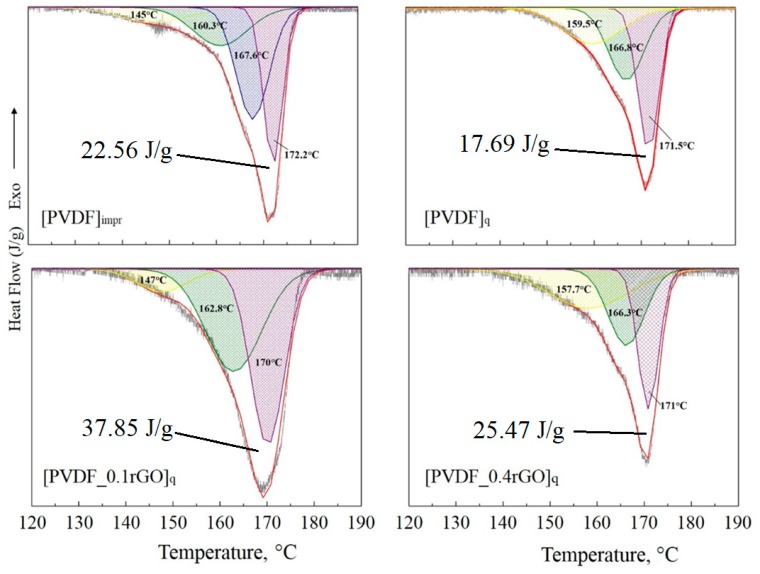
Differential scanning calorimetry (DSC) thermograms of the imprinted, quenched patterned PVDF films and the samples with 0.1 wt % and 0.4 wt % of rGO.

**Figure 7 polymers-11-01065-f007:**
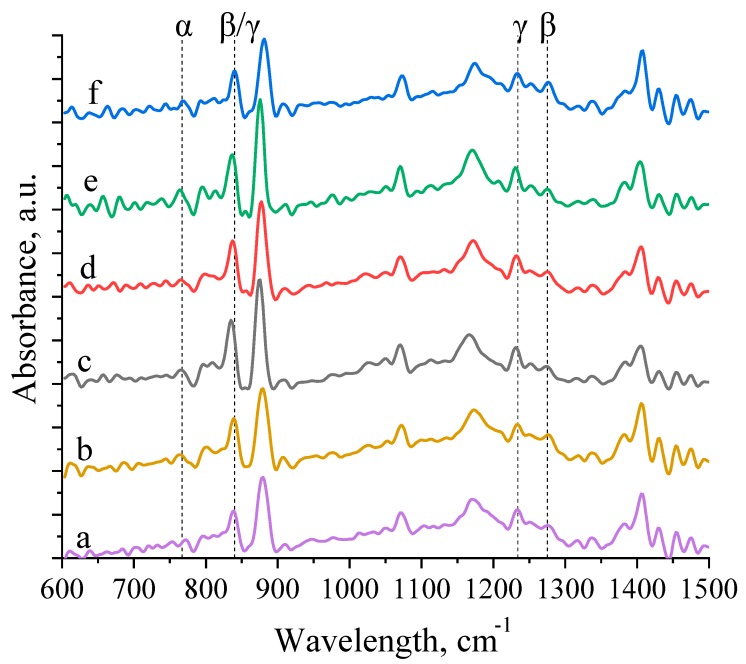
Fourier transform infrared spectroscopy (FTIR) spectra of (**a**) [PVDF]_impr_, (**b**) [PVDF]_q_, (**c**) [PVDF_0.1rGO]_q_, (**d**) [PVDF_0.1rGO]_p_, (**e**) [PVDF_0.4rGO]_q_, and (**f**) [PVDF_0.4rGO]_p_.

**Figure 8 polymers-11-01065-f008:**
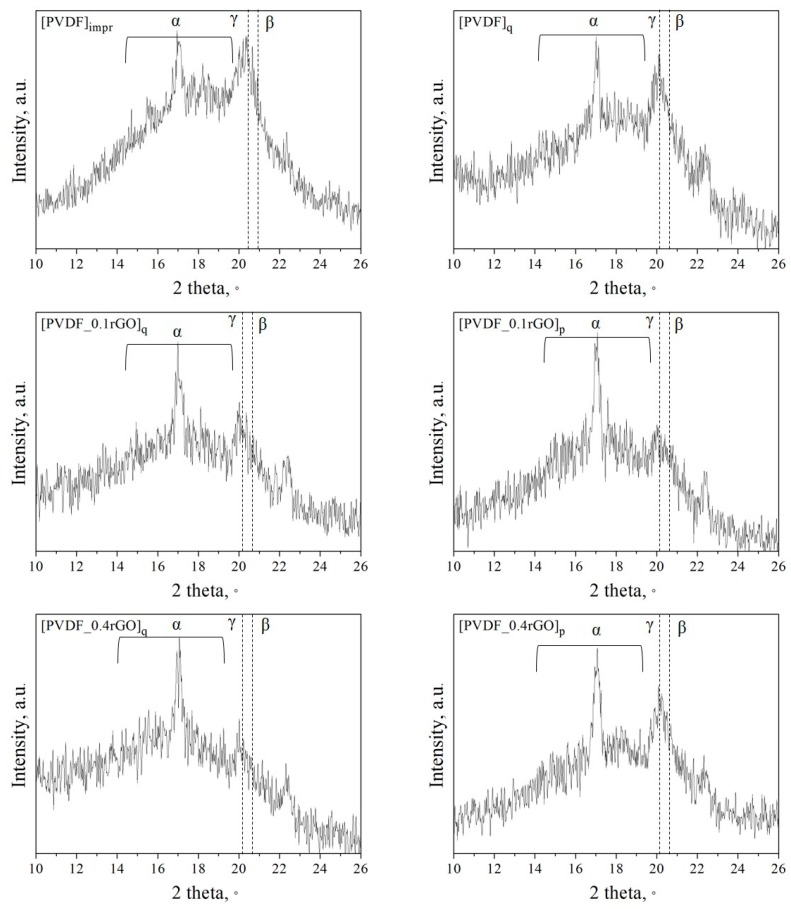
X-ray diffraction (XRD) of the imprinted and quenched PVDF micropillars and those filled with 0.1 wt % and 0.4 wt % rGO.

**Figure 9 polymers-11-01065-f009:**
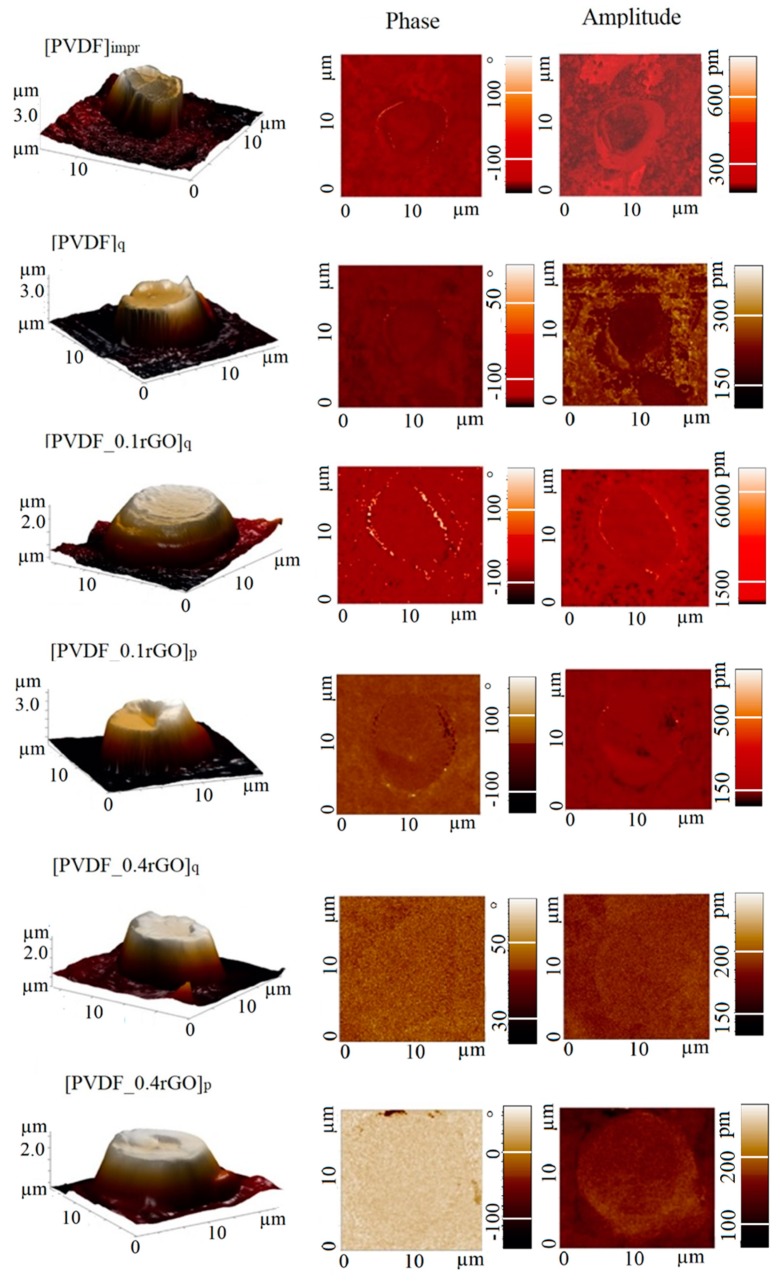
Three-dimensional topography and two-dimensional scans of the imprinted and quenched PVDF micropillars, and PVDF filled with 0.1 wt % and 0.4 wt % rGO.

**Figure 10 polymers-11-01065-f010:**
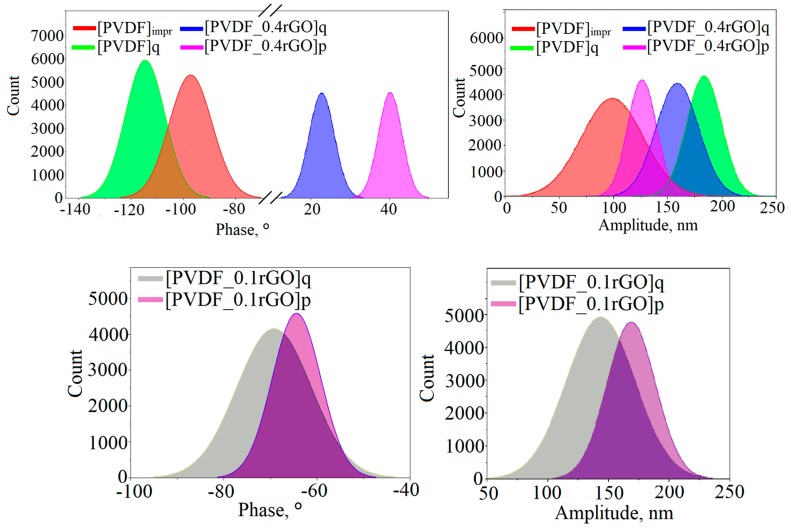
The statistical distribution of the PFM phase and amplitude values of the imprinted and quenched PVDF micropillars, micropillars with 0.4 wt % rGO (the measurements were carried out using an NSG03/TiN cantilever) and micropillars with 0.1 wt % rGO (FMG01 cantilever).

**Figure 11 polymers-11-01065-f011:**
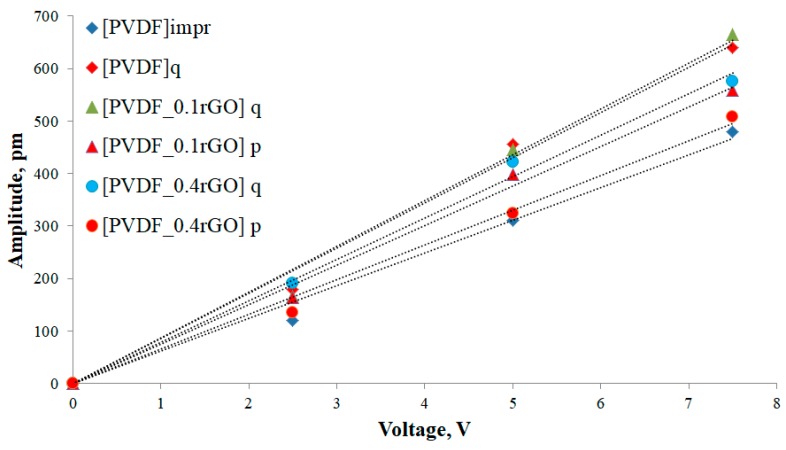
The measured piezoelectric signal, averaged over 4 different areas, versus the amplitude of the applied ac voltage.

**Table 1 polymers-11-01065-t001:** Description of the prepared samples.

No.	Group Label	Description
1	[PVDF]_impr_	Imprinted pristine PVDF micropillar array
2	[PVDF]_q_	Quenched pristine PVDF micropillar array
3	[PVDF_0.1rGO]_q_	Quenched composite PVDF/0.1 wt % rGO micropillar array
4	[PVDF_0.1rGO]_p_	Poled composite PVDF/0.1 wt % rGO micropillar array
5	[PVDF_0.4rGO]_q_	Quenched composite PVDF/0.4 wt % rGO micropillar array
6	[PVDF_0.4rGO]_p_	Poled composite PVDF/0.4 wt % rGO micropillar array

PVDF = poly(vinylidene fluoride), rGO = reduced graphene oxide.

**Table 2 polymers-11-01065-t002:** Crystallinity of the studied samples derived from the DSC data.

Sample	Crystallinity, %
[PVDF]_impr_	21
[PVDF]_q_	16
[PVDF_0.1rGO]_q_	36
[PVDF_0.4rGO]_q_	24

**Table 3 polymers-11-01065-t003:** Relative fraction of the α, β, and γ-phases estimated using FTIR spectra.

Sample	α, %	β, %	γ, %
[PVDF]_impr_	30(2)	16(7)	54(9)
[PVDF]_q_	30(4)	22(9)	48(10)
[PVDF_0.1rGO]_q_	22(3)	27(10)	51(16)
[PVDF_0.1rGO]_p_	26(1)	20(10)	54(3)
[PVDF_0.4rGO]_q_	32(9)	21(10)	47(18)
[PVDF_0.4rGO]_p_	32(10)	23(6)	45(4)

**Table 4 polymers-11-01065-t004:** Piezoelectric coefficient evaluated by averaging the PFM signal over 4 areas with dimensions of 20 × 20 μm in comparison with other published results.

Current Study		Literature Overview	
Sample	|d_33_^eff^|, pm/V	Sample	|d_33_^eff^|, pm/V (pC/N)
[PVDF]_impr_	64	Corona poled β-phase PVDF film [51]	28
[PVDF]_q_	86	Drawn PVDF film (stretching ratio = 5) [44]	34
[PVDF_0.1rGO]_q_	87	Poled PVDF film (E = 400 kV/cm) [50]	17.5
[PVDF_0.1rGO]_p_	75	Quenched at −20 °C PVDF film [16]	49.6
[PVDF_0.4rGO]_q_	79	γ-phase PVDF single nanoribbon [52]	58.5
[PVDF_0.4rGO]_p_	66	Simulated data for P(VDF-TrFE)/GO composite [53]	29.8

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
