# Peer review of "Piezoelectric Response in Hybrid Micropillar Arrays of Poly(Vinylidene Fluoride) and Reduced Graphene Oxide"

_polymers, 2019, doi:10.3390/polym11061065_

Reviewer 1 Report

The paper "Piezoelectric response in hybrid micropillar arrays of poly(vinylidene fluoride) and reduced graphene oxide" shows the results of the enhancement of the piezoelectric response of PVDF micropillar arrays. In my opinion the paper is quite well written and well prepared. It should be published in Polymers after small correction.

1/ The FTIR data could be collect in a table it will be usefull for readers

2/ Conclusion could be modify and  potential application should be more emphasis

Author Response

1 The FTIR data could be collect in a table it will be useful for readers

Answer:

The FTIR results are collected in the Table 3 in such a way to reveal different phases in the samples.

2 Conclusion could be modify and  potential application should be more emphasis

Answer:

The text was extended; a list of possible applications of the studied micropillars was added. The obtained micropillars investigated in this study can be used in different applications such as:

·         PVDF Actuator Configured for Sound Radiation Control-Smart Foam [1];

·         Sensor for Structural Health Monitoring [2];

·         Sensor for Ultrasonic Imaging [3];

·         High Energy Density Capacitor Material [4];

·         Touch or Tactile Sensor [5];

·         Energy Harvesting EEL [6];

·         Piezo-Actuator for Flapping Wing Micro Air Vehicles [7] etc.

Reviewer 2 Report

Comments to Authors:

1. Detailed information on fabrication and characterization parameters for the PVDF-rGO composite in experimental part should be added, such as the purity, company name and the reagent, accelerating voltage of the SEM, etc.

2. In Figure 6, the melting peak area of a samples should be given, and the change in crystallinity should be compared to explain the thermodynamic properties of the sample.

3. In Figure 11, it is hard to distinguish those dots since they are drawn so nearly, and the color contrast is not clear.

Author Response

1. Detailed information on fabrication and characterization parameters for the PVDF-rGO composite in experimental part should be added, such as the purity, company name and the reagent, accelerating voltage of the SEM, etc.

Answer:

The main parameters of the used chemicals were added to the text of the manuscript. In addition, the parameters of the FTIR, PFM and SEM were added in more details as well.

2. In Figure 6, the melting peak area of a samples should be given, and the change in crystallinity should be compared to explain the thermodynamic properties of the sample.

Answer:

According to the results of the study [8], the sample with 0.1 wt.% of rGO reveals better chemical resistance, stiffness, and strength than imprinted, quenched samples and the samples with 0.4 wt.% of rGO.

3. In Figure 11, it is hard to distinguish those dots since they are drawn so nearly, and the color contrast is not clear.

Answer:

The figure was changed to allow better distinguishing between different samples.

Reviewer 3 Report

The manuscript deals with the preparation of micropillar arrays of PVDF with varying amounts of reduced graphene oxide. These arrays were prepered by soft lithography and phae inversion at low temperature. rGO was included as a nucleating filler. The goal of the manuscript was to study the piezoelectric response of the hybrid materials obtained.

The manuscript is well written and the experimental part contains all details required. The characterization of the material is comprehensive using FTIR, XRD, DSC, SEM, PFM. The degree of crystallinity and more specifically the content of β phase material is analyzed. The piezoelectric response is investigated via PFM. The topic of the manuscript is important, for example with respect to sensingor sensor applications. Thus, the very good manuscript should be published in polymers.

I suggest a very few minor points that should be considered upon revision:

I like the fact that the DSC curves were deconvoluted in order to try to obtain more information. I am wondering why the authors did not deconvolute the IR spectra in order to get a better estimate of the β phase fraction. The method of using the absorbances relative to a “valley” just next to a peak appears to be less straight forward. If the spectra were deconvoluted in the interesting wavenumber range the fraction of β phase could be calculated from the resulting peaks. I do not want to suggest to redo the analyses. I am just wondering how the method chosen compares to the in my opinion more direct method.

Secondly, I noticed that the notation 40 +-2 pm/V. I guess using brackets around the numbers would be correct. Strictly speaking in the example given the unit refers only to the error.

Author Response

I suggest a very few minor points that should be considered upon revision:

I like the fact that the DSC curves were deconvoluted in order to try to obtain more information. I am wondering why the authors did not deconvolute the IR spectra in order to get a better estimate of the β phase fraction. The method of using the absorbances relative to a “valley” just next to a peak appears to be less straight forward. If the spectra were deconvoluted in the interesting wavenumber range the fraction of β phase could be calculated from the resulting peaks. I do not want to suggest to redo the analyses. I am just wondering how the method chosen compares to the in my opinion more direct method.

Answer:

IR-spectra were processed using the Fourier Self-Deconvolution add-on for the OriginPro 2018. After that, the total amount of the electroactive phase was calculated, but the specific values of the β and γ-phases were determined with the method already used before and described in the study [9]. All the changes are reflected in Figure 7 and Table 3.

According to the updated calculations, the samples with 0.1 wt.% of rGO reveal the largest amount of β-phase, while the samples with 0.1 wt.% of rGO have the least amount of α-phase compared to all the studied samples. In addition, the samples with 0.1 wt.% of rGO have the highest degree of crystallinity among all the studied samples.

Secondly, I noticed that the notation 40 +-2 pm/V. I guess using brackets around the numbers would be correct. Strictly speaking in the example given the unit refers only to the error.

Answer:

The requested corrections were done in the text of the manuscript.